# Virtual lesions in MEG reveal increasing vulnerability of the language network from early childhood through adolescence

Brady J. Williamson [1], Hansel M. Greiner[2,3] & Darren S. Kadis [4,5] ✉

In childhood, language outcomes following brain injury are inversely related to age. Neuroimaging findings suggest that extensive representation and/or topological redundancy may confer the pediatric advantage. Here, we assess whole brain and language network resilience using in silico attacks, for 85 children participating in a magnetoencephalography (MEG) study. Nodes are targeted based on eigenvector centrality, betweenness centrality, or at random. The size of each connected component is assessed after iterated node removal; the percolation point, or moment of dis-integration, is defined as the first instance where the second largest component peaks in size. To overcome known effects of fixed thresholding on subsequent graph and resilience analyses, we study percolation across all possible network densities, within a Functional Data Analysis (FDA) framework. We observe age-related increases in vulnerability for random and betweenness centrality-based attacks for whole-brain and stories networks (adjusted-$p < 0.05$). Here we show that changes in topology underlie increasing language network vulnerability in development.

In childhood, functional outcomes following brain injury often have an inverse relationship with age, a phenomenon frequently referred to as the Kennard Principle (see, however[1]). In the domain of langauge, the "pediatric advantage" is indeed well-established[2–4]. Neuroimaging has shown that language is supported by a bilateral and diffuse network in early childhood, which becomes increasingly left lateralized and focal through adolescence[5–9]. The neuroimaging data suggest that either extensive representation and/or topological redundancies confer the advantage[7].

Redundancy is a ubiquitous trait in neural circuits and allows for flexible adaptations both in health and disease[10]. Redundancies may provide network robustness to neurological and experiential perturbations, especially early in development[11]. Brain network analysis provides a framework in which to investigate topological redundancy. Networks can be represented as graphs based on structural and/or functional connections[12]. In silico attacks, involving targeted and iterated removal of regions within networks can be used to simulate lesion impact, and to assay robustness of the network. Robustness metrics traditionally relate the size of the largest connected component within a network to all nodes/regions available[13]. The point at which the "lesioning", or removal of regions leads to disintegration, or fracture, of the largest connected component so that its size is 0, is defined as the *percolation point*. However, there are both theoretical and technical limitations of using this definition of percolation point in finite systems, such as brain networks. Specifically, in finite systems, the size of the largest connected component only reaches 0 when all nodes are removed, meaning that there is no general/useful criterion for determining the percolation point from the size of the largest connected component[14]. An alternative that avoids this limitation and captures the redundancy of neural circuits is using the second largest connected component of the network that may subsume some of the workload if there is damage to the system[14].

[1]Department of Radiology, University of Cincinnati, Cincinnati, OH, USA. [2]Division of Neurology, Cincinnati Children's Hospital Medical Center, Cincinnati, OH, USA. [3]Department of Pediatrics, College of Medicine, University of Cincinnati, Cincinnati, OH, USA. [4]Neurosciences and Mental Health, Hospital for Sick Children, Toronto, ON, Canada. [5]Department of Physiology, University of Toronto, Toronto, ON, Canada. ✉e-mail: darren.kadis@sickkids.ca

Two primary strategies can be used when assessing the effects of in silico attacks on a brain network: random and targeted. For targeted attacks, there are several possible metrics that can be used to rank nodes in order of removal. The two strategies used in this work are ranking nodes based on eigenvector centrality (EC) and betweenness centrality (BC), metrics that assess the importance of a node for network functioning[15]. EC captures the relative popularity of a node, and can be thought of as an extension of degree, which is simply the number of connections a node possesses. Eigencentrality (and related metrics, such as PageRank), consider both the number of connections a node has, as well as the number of connections its topological neighbours possess[12]. The advantage of eigenvector centrality over other higher-order metrics of importance based on popularity, is that the metric is derived directly from the eigenvalue of the adjacency matrix, requiring no tuning (cf PageRank, which requires setting 'damping factors' dependent on various assumptions). BC reflects the critical positioning of a node within a network. Nodes with high betweenness centrality serve as bridges, allowing for informational flow to/from other nodes.

When performing brain network analyses, a ubiquitous problem with no clear-cut solution is that of optimal thresholding. Certain topological parameters of brain connectivity are connection-density dependent, making selection of an optimal density nontrivial[16]. Typically, a constant threshold based on a metric of connection strength is applied to brain graphs to reduce computational complexity and to eliminate the weakest connections[17]. Additionally, researchers have attempted to apply data-driven strategies, such as those based on Minimum Spanning Trees (MSTs) or percolation analysis[16,18]. However, these strategies still necessitate selection of an initial density for each participant to calculate metrics that will be analyzed with group statistics. An alternative would be to fit a statistical model at several common densities, independently, but this approach leads to the problem of multiple comparisons, that if not corrected for, leads to an inflated Type I error rate; if corrected for, investigtors may face reduced power (increased Type II error).

One possible framework for overcoming this limitation is functional data analysis (FDA)[19]. As opposed to models that accommodate single scalars as independent predictors of a response variable, FDA allows for the modeling of functions as independent and/or dependent variables. Applied to the current aims, network metrics can be modeled across a broad range of initial densities to generate a function for each study participant. The functions are then used in multivariate statistical models. The technique overcomes the arbitrary selection of initial graph density for subsequent attack analyses (i.e., node-based attacks on network in silico), as well as multiple comparison problems inherent in mass univariate approaches.

The current study sought to determine differences in percolation point during typical development in a large cohort of children, 4 to less than 19 years of age, using an FDA framework in which initial network density is parameterized. We hypothesized that there would be an inverse relationship between percolation point and age, indicating greater robustness in the functional brain networks of young children, consistent with literature on the pediatric advantage. Two important contributions of this work include: utilization of percolation analysis to assess brain network resilience in development, and implementation of an analysis framework that circumvents the need to preselect network thresholdings.

## Results
### Whole-brain results

Analyses revealed overall model significance for random attacks ($F = 5.99$, $p < 0.0001$, $R^2 = 0.947$, functional $R^2 = 1.3\%$). There was a significant effect of age while controlling for sex, handedness, and mean node distance on percolation point determined by random attacks at densities below 15% ($F = 31.44$, $p < 0.0001$). Bootstrapped betas for age show a negative relationship (Fig. 1), where percolation point decreases with age, indicating younger children's networks are more robust to failure. There was no significant effect of sex or handedness while controlling for age and mean node distance in this model. Results for random attacks for the whole-brain parcellation are summarized in Fig. 1, Supplementary Table 1, and Supplementary Fig. 3.

We observed a significant model for the BC-based attack strategy ($F = 6.82$, $p < 0.0001$, $R^2 = 0.934$, functional $R^2 = 8.20\%$). There was a significant effect of age ($F = 126.38$, $p < 0.0001$). Across all densities, bootstrapped betas showed a negative relationship between age and percolation point (Fig. 2). There was no significant effect of sex or

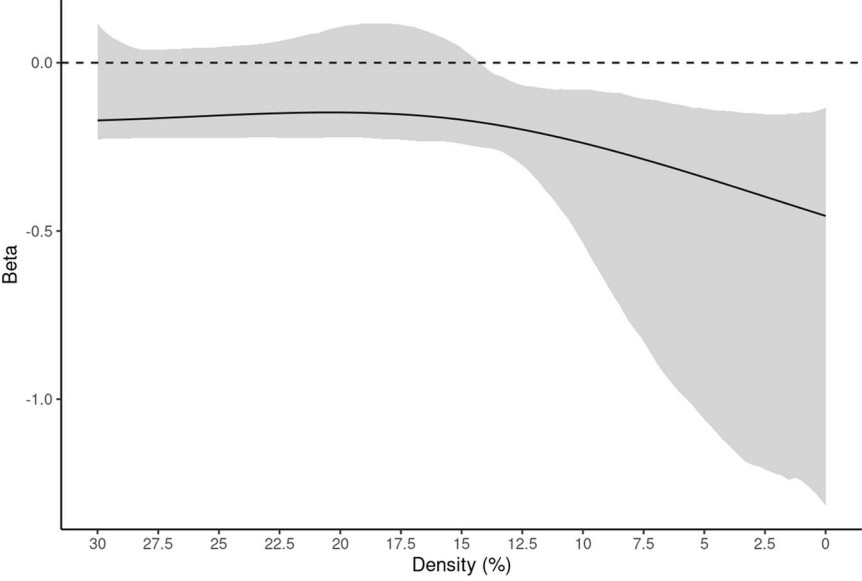

**Fig. 1 | Results for the Random attack strategy on the whole-brain parcellation.** Data are presented as the beta estimate across densities +/− 95% CIs (shaded). After testing for significance of the overall model (see, Supplementary Table 1), analyses showed Age (negative) as the only significant regressor at all densities below 15% ($F = 31.44$, $p < 0.0001$). Sex and Handedness did not meet the $p$-value threshold (0.01) and 95% CIs contained 0 throughout the whole density range (Supplementary Fig. 3, panels c and d). Source data are provided as a Source Data file.

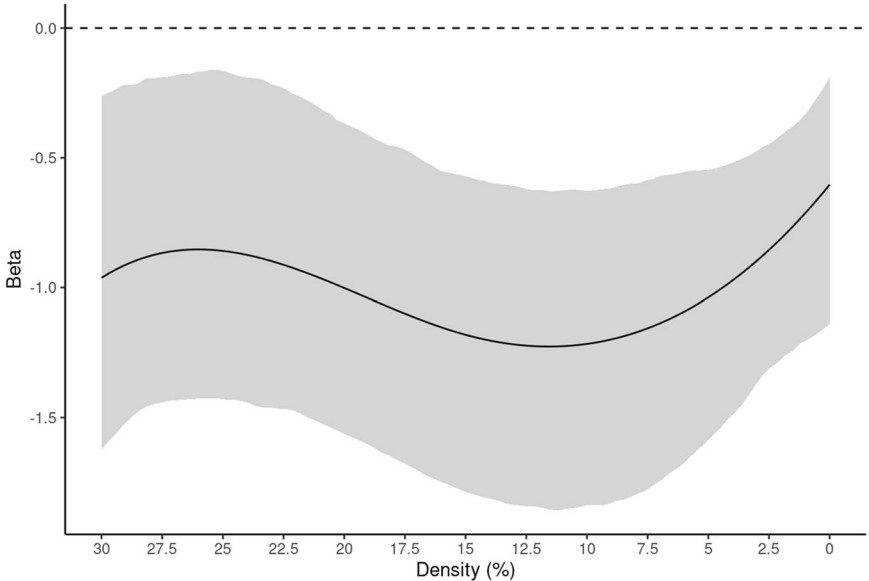

**Fig. 2 | Results for betweenness centrality-based attacks on the whole-brain parcellation.** Data are presented as the beta estimate across densities +/− 95% CIs (shaded). After testing for significance of the overall model (see, Supplementary Table 1), analyses showed Age (negative) as the only significant regressor at all densities ($F = 126.38$, $p < 0.0001$). Sex and Handedness did not meet the $p$-value threshold (0.01) and 95% CIs contained 0 throughout the whole density range (Supplementary Fig. 4, panels c and d). Source data are provided as a Source Data file.

handedness while controlling for age and mean node distance in this model. Results for BC-based attacks for the whole-brain parcellation are summarized in Fig. 2, Supplementary Table 1, and Supplementary Fig. 4. There were no significant effects of age, sex, or handedness on percolation point at any densities for EC based attacks (Supplementary Fig. 5).

To visualize the effects seen in the FDA results, we plotted regions that were removed before network failure at 5% density, colored according to frequency of removal across all participants. This density was chosen as it was included in all analyses in which there was a significant effect of age and provided enough sparsity for reasonable interpretation of results. At higher densities, the node removal maps became oversaturated, and at lower densities, there was not enough variation among regions, making results difficult to interpret. To compare across ages, we performed this visualization for the 1st age quartile and 4th age quartile separately. Visualization of these "group-level" hubs based for BC-based attacks showed a pattern in which younger participants had more regions that were removed consistently across subjects, i.e., goup-level hubs, at the point of network failure (brighter regions in panel a of Fig. 3, panel a) compared to older participants (Fig. 3, panel b). The younger participants had more distributed posterior (occipital/parietal) critical hubs, which may reflect relative reliance on visualization strategies for the language tasks; however, this has not been tested, experimentally. The distribution of critical hubs becomes more focal in the older group as evidenced by greater heterogeneity among neighboring regions. To show consistency across different densities, plots at 2.5% (Supplementary Fig. 9, left panel) and 7.5% densities were also generated (Supplementary Fig. 10, left panel).

**Stories Network Results**
Analyses showed a significant overall model effect for random attacks within the stories network ($F = 13.39$, $p < 0.0001$, $R^2 = 0.965$, functional $R^2 = 2.7\%$). Modeling revealed a significant effect of age ($F = 44.51$, $p < 0.0001$). Bootstrapped betas showed a negative effect of age from 1-15% density (Fig. 4). Results for random attacks on the stories network parcellation are summarized in Fig. 4, Supplementary Table 1, and Supplementary Fig. 6.

There was also a significant overall model effect for the BC-based attack strategy ($F = 6.75$, $p < 0.0001$, $R^2 = 0.915$, functional $R^2 = 5.7\%$). The model again showed a significant effect of age ($F = 86.25$, $p < 0.0001$). Bootstrapped betas showed a negative effect of age across all densities (Fig. 5). Results for BC-based attacks on the stories network parcellation are summarized in Fig. 5, Supplementary Table 1, and Supplementary Fig. 7. There were no significant effects of age, sex, or handedness on percolation point at any densities for EC based attacks (Supplementary Fig. 8).

Using the same visualization technique as in the whole-brain results, we found that the younger children again had more regions that were removed consistently, i.e., goup-level hubs, across subjects (brightest regions of Fig. 6, panel a). Older participants had fewer of these regions, (Fig. 6, panel b). Similar to the whole-brain results, plots at 2.5% (Supplementary Fig. 9, right panel) and 7.5% (Supplementary Fig. 10, right panel) densities showed broad consistency with the maps at 5% density.

## Discussion
The current study sought to investigate the effects of age on brain network robustness by utilizing a density-independent framework to avoid problems inherent with network thresholding. Our results suggest increased network vulnerability from childhood through adolescence, which supports prior literature on the pediatric advantage[2–4].

An interesting aspect of the current study is that the effect of age on network vulnerability was observed for random and BC-based attacks, but not EC-based attacks. The most robust effect was seen with betweenness centrality, a measure of nodal importance based on the amount of influence the node has on the information flow through the network. Though EC-based attacks were the most effective for network dismantling, with an earlier percolation point than BC-based attacks (Supplementary Figs. 1 and 2), our results suggest EC-based attacks are equally effective across childhood. The distinction between nodes with high BC versus those with high EC is largely consistent with the notion of connector versus provincial hubs[20], though the delineation may be more appropriately characterized as domain-specific and domain-general connector hubs. In functional brain networks, domain-general regions facilitate long-range connections between domain-specific

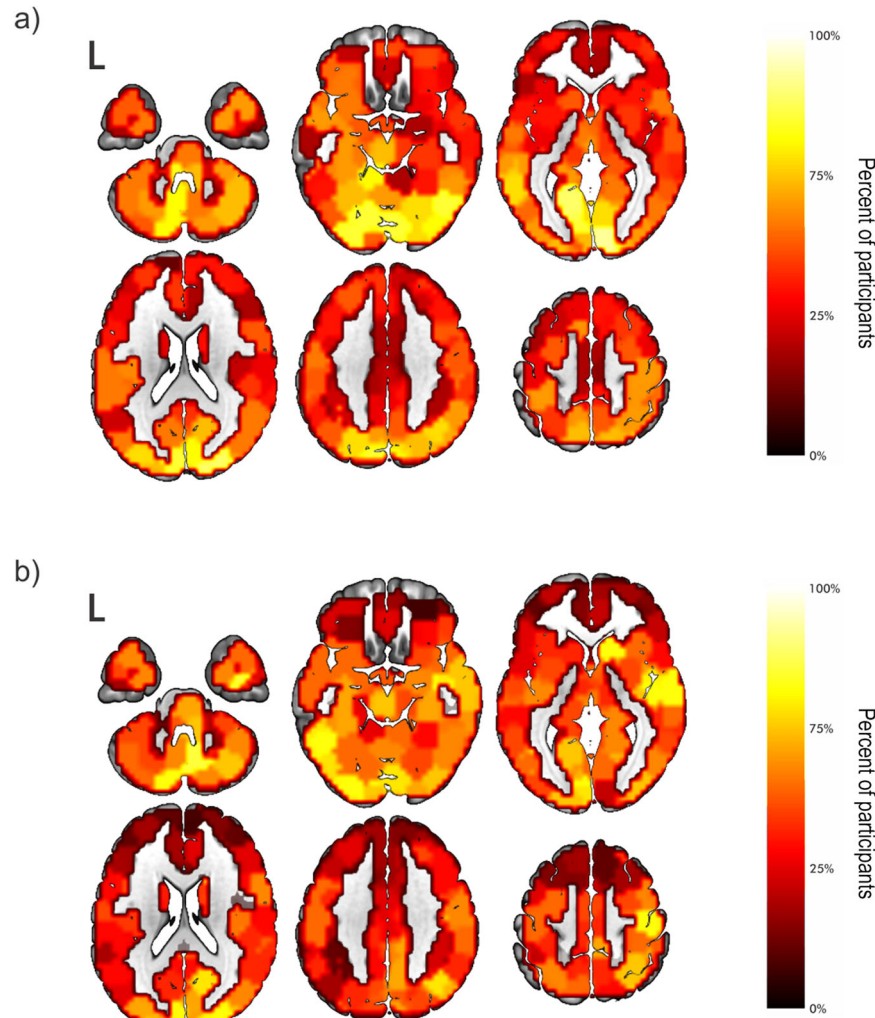

**Fig. 3 | Distribution of nodes removed prior to failure in whole-brain attacks.**
Distribution of nodes removed in the whole-brain for betweenness centrality-based attacks for the 1st (panel **a**) and 4th (panel **b**) quartiles of age. Regions are colored by the percentage of participants for which the node was removed prior to network failure (dark to light). These results were displayed for 5% density across participants. Results show more consistently removed regions, i.e., group level hubs, in the younger quartile (brighter regions in panel **a**) compared to older participants (panel **b**). Also, distribution of critical hubs becomes much more focal in the older group as evidenced by greater heterogeneity among neighboring regions. To show consistency across densities, results are also plotted at 2.5% (Supplementary Fig. 9, left panel) and 7.5% (Supplementary Fig. 10, left panel) density.

clusters (also called "convergence zones"[21]). Since the local hubs of these domain-specific clusters must go through domain-general regions, their eigenvector centrality is high (i.e., domain-general hubs are connected to many domain-specific hubs). Regions within the language network with high betweenness centrality are most likely involved in domain-specific functioning, as function-specific information is passed through these regions from local functional clusters to the primary coordinating areas (i.e., provincial/domain-general hubs). Further, researchers have found a key aspect of brain maturation is the pruning of short range connections and an increase of long-range connections that facilitate efficient information transfer between distant regions, leading to an overall increase in characteristic path length[22].

This, together with our current findings and previous literature on the pediatric advantage, suggests the increase of efficiency resulting from the tradeoff between short- and long-range connections during development comes with increased network vulnerability. Since BC-based attacks were more effective for network dismantling in older participants and EC-based attacks were equally effective for all ages, it seems domain-specific hubs become increasingly important while domain-general connector hubs remain stable across development,

consistent with the increase in long-range "expensive", and seemingly more vulnerable, connections. Our results within the language network, specifically regarding participants in the 4th age quartile, are consistent with our previous work on connector hubs in the language network, defined by regions most involved in interfrequency communication between regions[23,24].

The current findings have both clinical and theoretical implications. Clinically, arguments have been recently made that we should reconsider traditional approaches to surgical resection (i.e., functional localization) in favor of a network-based approach[25]. Two of the primary features that can be used in this framework to guide surgical decision making are: (1) whether there is tissue available that can subsume functional load after resection (i.e., the plastic potential) and (2) what are the downstream effects on the network of removing a particular piece of cortex (i.e., potential for dismantling). This is especially relevant in children as the language network is still bilateral and distributed, becoming increasingly left-lateralized and focal with age[7,9]. It is crucial to understand the specific role each piece of cortex is playing in the larger network at a particular stage of development and the capacity of other cortical areas to compensate for injury to that node before performing resection to properly assess the risks of surgery.

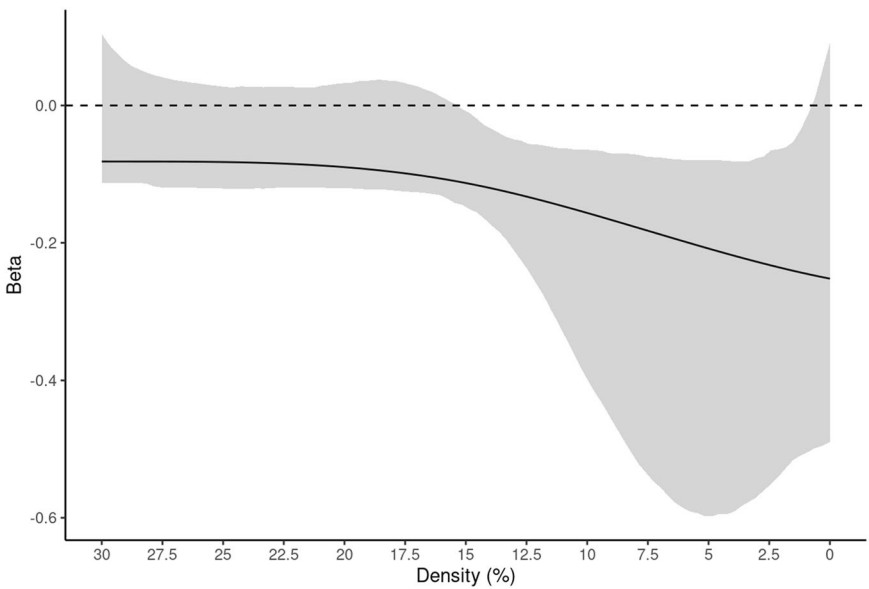

**Fig. 4 | Results for random attacks on the stories network parcellation.** Data are presented as the beta estimate across densities +/− 95% CIs (shaded). After testing for significance of the overall model (see, Supplementary Table 1), analyses showed Age (negative) as the only significant regressor at densities between 1 and 15% ($F = 44.51$, $p < 0.0001$). Sex and Handedness did not meet the $p$-value threshold (0.01) and 95% CIs contained 0 throughout the whole density range (Supplementary Fig. 6, panels c and d). Source data are provided as a Source Data file.

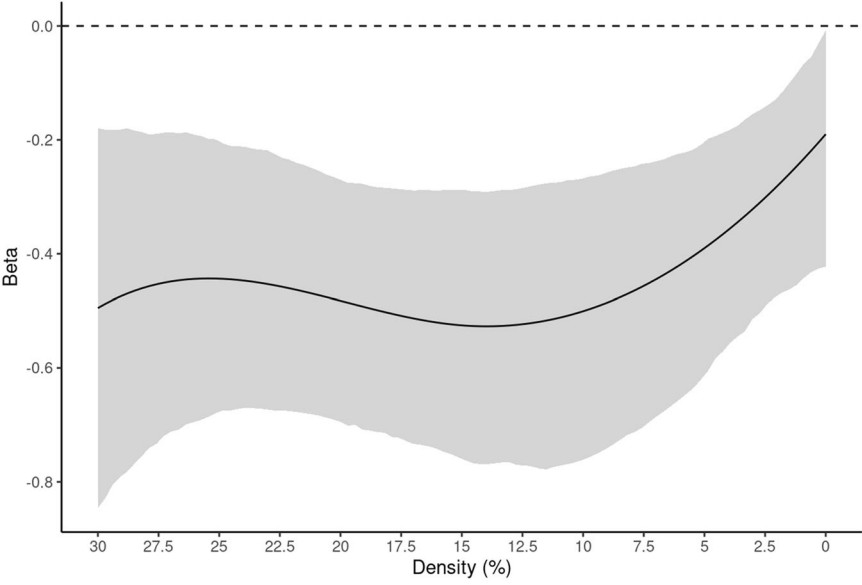

**Fig. 5 | Results for betweenness centrality-based attacks on the stories network parcellation.** Data are presented as the beta estimate across densities +/− 95% CIs (shaded). After testing for significance of the overall model (see, Supplementary Table 1), analyses showed Age (negative) as the only significant regressor at all densities ($F = 86.25$, $p < 0.0001$). Sex and Handedness 95% CIs contained 0 throughout the whole density range (Supplementary Fig. 7, panels c and d). Source data are provided as a Source Data file.

Theoretically, our findings support the notion that there exists both domain-specific and domain-general hubs that should be considered differently when assessing potential for residual tissue to support function[24]. The hubs that, when removed, led to dismantling of the network most often in our youngest quartile were bilateral and consistent across participants, especially compared to adjacent tissue (Fig. 6, panel a, brightest regions). There is a clear shift in the oldest quartile in which these regions are not as consistent, but more adjacent regions have comparable importance (Fig. 6, panel b). Our results for these domain-specific hubs are consistent with previous findings that language can reorganize both inter- and intra-hemispherically and is more likely to reorganize intrahemispherically with age[26,27]. If we maintain the hypothesis that greater importance, determined by the current methods, suggests capacity to subsume functional load, then these results lead to the conclusion that there are more bilateral critical sites in young children that can compensate if there is injury. In contrast, the "spread" of importance from strong hubs to adjacent regions in the older participants is consistent with intra-hemispheric compensation[26] and inconsistency of recovery, since the remaining hubs no longer have the capacity to subsume the functional load. Future studies will aim to use these insights to design analytic pipelines for mapping critical hubs for specific functions at the single subject level.

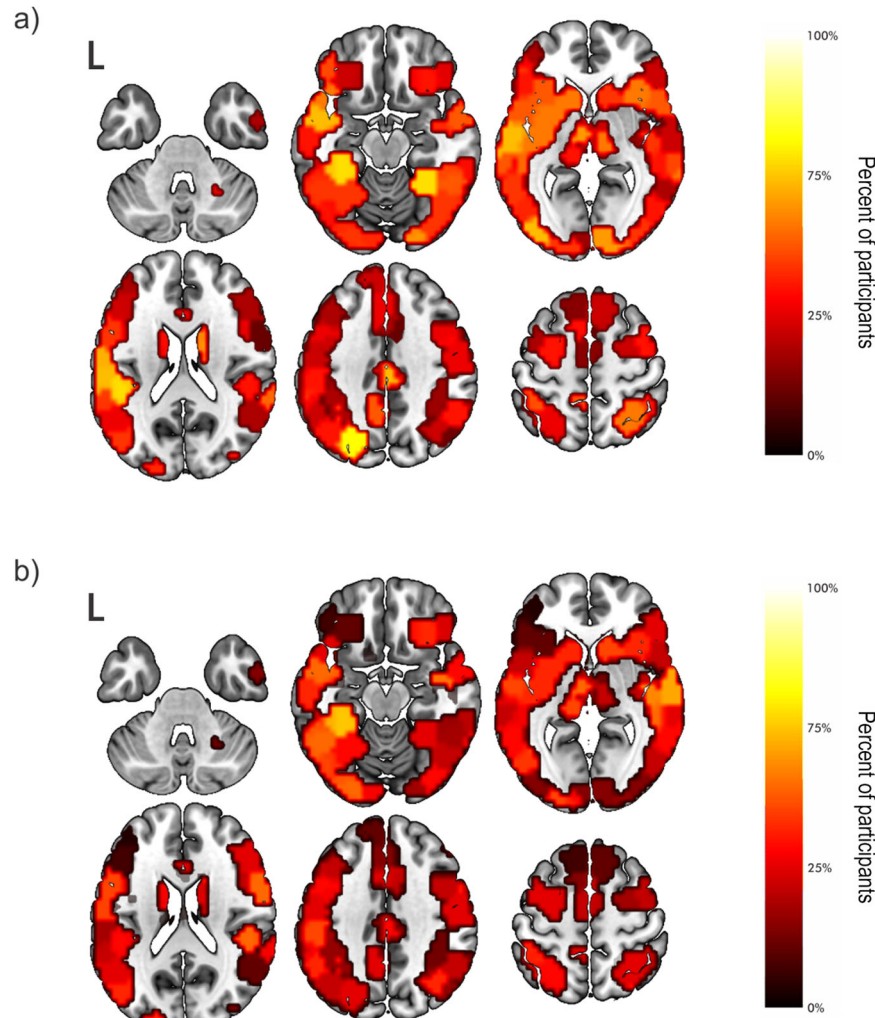

**Fig. 6 | Distribution of nodes removed prior to failure in stories network attacks.** Distribution of nodes removed in the stories network for betweenness centrality-based attacks for the 1st (**a**) and 4th (**b**) quartiles of age. Regions are colored by the percentage of participants for which the node was removed prior to network failure (dark to light). These results were displayed for 5% density across participants. Like whole-brain analyses, we found that the younger children had more regions that were removed consistently, i.e., group-level hubs, across subjects (panel a). To show consistency across densities, results are also plotted at 2.5% (Supplementary Fig. 9, panels c and d) and 7.5% (Supplementary Fig. 10, panels c and d) density.

## Methods

### Participants

Eighty-five typically-developing children and adolescents, ages 4 to less than 19 years, were recruited for MEG language studies at Cincinnati Children's Hospital Medical Center (CCHMC) between 2014 and 2020. A total of 82 participants (45 female), ages 4 years 0 months to 18 years 7 months, provided high-quality MEG and MRI data that contributed to the current analyses (Table 1). All participants were native English speakers, free from history of neurological or hearing impairment, speech- or language deficits, and learning disability, as per family report. Informed written consent from a parent or legal guardian was obtained for all participants less than 18 years of age; participants between the ages of 10 and 18 years also provided assent. Participants older than 18 years provided informed written consent. The study was approved by the Institutional Review Board at CCHMC (data collection site), and the Research Ethics Board at the Hospital for Sick Children in Toronto.

### Data acquisition

**MEG stories listening.** Data were acquired on a 275-channel whole-head MEG system (CTF MEG Neuro Innovations, Inc., Coquitlam, BC, Canada; Acq 5.4.2 software), at 1200 Hz. All participants were studied in the supine position, with memory foam pads and/or linens cushioning the head to promote comfort and stability. Head position was monitored continuously via localization coils placed over nasion and preauricular locations. The stories listening paradigm has been used extensively (e.g., MEG[28] fMRI[29]), and is described only briefly, here. Participants listened to child-friendly stories, read aloud by a female-speaker in sentences 2-3 seconds in duration (stories trials). Alternately, children listed to speech-shaped noise, matched for spectral content and amplitude envelope, of identical duration (noise trails). A total of 48 stories, and 48 noise trials, were presented. Stimuli were delivered binaurally via a calibrated audio system, comprised of distal transducers, flexible tubing, and disposable foam insert earphones (Etymotic Research, Inc., IL, USA).

**Structural MRI.** In all cases, MRI was acquired after MEG. Radiopaque markers were placed over the MEG fiducial positions prior to scanning, permitting precise coregistration across modalities. 3D T1-weighted images (1.0 × 1.0 × 1.0 mm voxels, MDEFT sequence) were acquired at 3.0 T for all participants, on either a Philips Achieva or Philips Ingenia Elition scanner (Philips Medical Systems, International; Philips MR release 5.1/5.6).

**Table 1 | Age, Sex, and Handedness of all participants, by age quartile**

|  | First (*n* = 21) | Second (*n* = 20) | Third (*n* = 20) | Fourth (*n* = 21) | Overall (*n* = 82) |
|---|---|---|---|---|---|
| **Age** |  |  |  |  |  |
| Mean (SD) | 5.46 (0.69) | 8.41 (1.77) | 13.1 (1.07) | 17.0 (0.80) | 11.0 (4.61) |
| Median [min, max] | 5.67 [4.0, 6.4] | 7.00 [6.59, 11.0] | 13.0 [12.0, 15.0] | 17.0 [16.0, 18.7] | 11.5 [4.0, 18.7] |
| **Sex** |  |  |  |  |  |
| Male | 6 (28.6%) | 11 (55.0%) | 11 (55.0%) | 9 (42.9%) | 37 (45.1%) |
| Female | 15 (71.4%) | 9 (45.0%) | 9 (45.0%) | 12 (57.1%) | 45 (54.9%) |
| **Handedness** |  |  |  |  |  |
| Left | 1 (4.8%) | 1 (5.0%) | 1 (5.0%) | 0 (0%) | 3 (3.7%) |
| No Preference | 3 (14.3%) | 3 (15.0%) | 1 (5.0%) | 0 (0%) | 7 (8.5%) |
| Right | 17 (81.0%) | 16 (80.0%) | 18 (90.0%) | 21 (100%) | 72 (87.8%) |

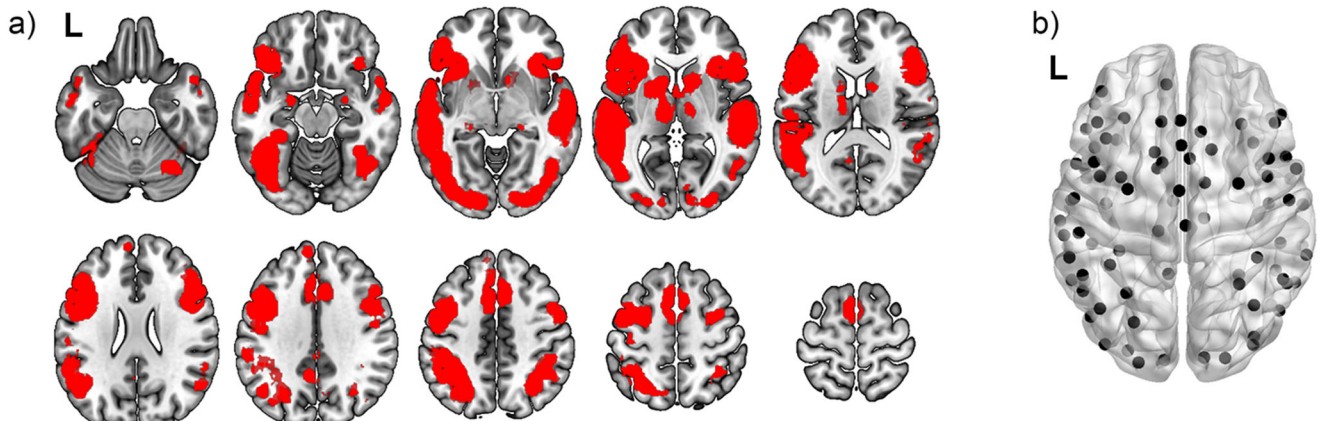

**Fig. 7 | Distribution of language network nodes.** Nodes were derived from NeuroSynth by using a uniformity test (*FDRq* < 0.01) on all resulting activation maps with the search term "language". Panel **a** represents the volumetric regions contained in the parcellation. Panel **b** is a 3D-rendered representation showing the centroid of each node.

## Defining Whole Brain and Language Networks

Node positions were specified as the centroids of a ~ 200-unit random parcellation scheme[30]; the parcellation yields 194 nodes for the whole brain network. The language network was defined through an automated neuroimaging meta-analytic tool, NeuroSynth (neurosynth.org[31]), with the search term "language" (May 1, 2022 query). The resulting map, based on 1101 studies, represents fMRI activation across a broad spectrum of research related to language, including developmental studies. NeuroSynth can output maps based on an *association test* (activations that are specifically related to the search term), or a *uniformity test* (activations that co-occur with the search term). We preferred the uniformity map (FDRq = 0.01), which was relatively extensive, and included bilateral perisylvian regions known to support language in early childhood. The language map was binarized, then assessed for overlap with the parcellation image; parcels with at least 10% activation occupancy (89/194 parcels; 52 in the left hemisphere) were included in the language network (Fig. 7).

## MEG Preprocessing and Source Localization

Preprocessing and source localization were carried out using FieldTrip[32] routines running in MATLAB (v 9.10.0; The Mathworks, MA, USA). Continuous recordings were initially bandpass filtered from 0.1 to 100 Hz, and power line noise (60 Hz) was suppressed via a narrow notch filter. Filtered continuous data were then subjected to independent component analysis (ICA), and the spatio-temporal characteristics of the first 20 components, ordered by variance, were visually assessed; stereotypical ocular and cardiac artifact components (0-9 components per subject, mean = 3.82) were identified and rejected, prior to projection back to sensor space. The data were then epoched 0-2000 ms relative to the onset of each sentence (stories trials) or speech-shaped noise (noise trials).

Realistic single-shell sourcemodels were constructed from segmented 3D T1 weighted images[33]. Network node positions were non-linearly warped from template to individual subject space using SPM12 (https://www.fil.ion.ucl.ac.uk/spm/) routines. Stories and noise trials were concatenated for covariance estimation, and construction of a common spatial filter; activity at each location estimated via a linearly consttrained minimum variance beamformer[34], with 0.1% regularization. Source activity for stories listening trials, only, were used in functional connectivity and subsequent attack analsyes.

## Functional Connectivity

Recently, we have shown that connectivity patterns differ across frequencies, in children completing language tasks in MEG (Kadis et al., 2016; Sharma et al., 2022); likewise, we have seen spectrally-focused connectivity differences in several patient populations completing language tasks in EEG and MEG (e.g., Barnes-Davis et al., 2018, 2021a, 2021b; Farah et al., 2019). To retain sensitivity to spectrally-focused effects, we assess connectivity in narrow frequency bins, rather than within canonical bands or broadband. Trial-wise Fourier representations were obtained for 0.5-100 Hz signal in 0.5 Hz steps with ±2 Hz smoothing via discrete prolate Slepian sequence multitapers. Pair-wise functional connectivity was assessed using weighted phase lag index (wPLI[35]) for each frequency bin, and aggregated in the L2 norm (Euclidian distance) to estimate total coupling across the spectra. Adjacency matrices were established for whole brain and language

networks. To control for potential differences related to field spread effects on small (young) versus larger (adolescent) brains[36], mean Euclidean distance was calculated for all node pairs, and included as a covariate in statistical models.

## Attack Analyses

Whole brain and language network attack analyses were carried out separately. Networks were proportionally thresholded, with densities ranging from 100% to 0.25%, in 0.25% steps. At each density, we removed nodes at random, or removed nodes based on their centrality (eigenvector, betweenness) values in sequential order from those nodes with highest centrality to those with the lowest. The Brain Connectivity Toolbox (BCT) was used to compute all graph metrics[12].With each node removal, we compute the size of the largest and second largest connected components. The percolation point was defined as the fraction of nodes removed for which the size of the second largest connected component peaked in size[14]. The attacks were iterated 100 times, to establish robust percolation point estimates (essential for the random approach; to account for possible ties in the targeted attacks, we randomly select among the set of nodes sharing maximal centrality). Mean percolation point for each density was recorded.

## Functional data analysis - preprocessing

To prepare the data for multivariate modeling, data were preprocessed to identify and remove outliers. First, participants that had a "spike" of percolation point at the lowest density (0.25%), likely due to instability in the percolation point calculation when there are so few nodes, were removed by extracting the last value of the function for all participants, calculating the z-scores for these values, and removing any participant for which this last value had a z-score > 2. Next, percolation point by density functions were calculated using a $4^{th}$ order b-spline with 402 (number_of_densities+order-2) basis functions. First, these initial functions were trimmed based on the first derivative of the mean group function to reduce computational requirements. Since we were interested in the point (i.e., number of nodes removed) at which group differences were most apparent, we focused on the precipitous decline in the generated functions where percolation point varied from a static range of values. To calculate this point, we identified the initial network density at which the standard deviation of the first derivative (i.e., velocity) of the function became negative.

Each individual function was then smoothed by term $\lambda$, that optimally penalizes the roughness of the second derivative, determined empirically by testing an exponential range of options ($e^{-5-12}$) then finding the value that minimizes generalized cross validation (GCV)[19,23]. The sum of square residuals was calculated for each smoothed function and used to determine outliers. In this case, outliers represented poor fits to the smoothed function on an individual basis.

## Functional data analysis—modeling

Function-on-scalar (FoSR) regression[19,37], specifically the penalized flexible functional regression approach (PFFR[38,39,]), was performed to determine the effects of age, sex, and handedness on percolation point by density. Mean Euclidean node distance (See Methods−Functional Connectivity) was included as a nuisance regressor in all models. Previous work suggests "raw", or unsmoothed, functions should be used in modeling because pre-smoothing the data eliminates potentially important variability and measurement error in the functional responses[38]. Following this recommendation, we used the unsmoothed functions as the dependent variable (DV) in our model and let the correct smoothing parameters be chosen during the regression. Because the functions are still smoothed/penalized during the regression, the smoothing and related outlier detection during

preprocessing is still important to remove ill-fitting functions prior to modeling.

Beta estimates for each independent variable (IV) were allowed to vary across densities and smoothed with the same parameters as the DV. All variables in the model were smoothed using 5 cubic b-splines with a first order difference penalty (i.e., P-splines[39]). Model assumptions were checked using Q-Q, Residual vs Regressor, and Response vs Fitted plots, as well as assessing the distribution of the model residuals. K-basis dimension checking was performed to ensure the number of splines used for modeling was adequate[40].

## Functional data analysis−hypothesis testing

Because FoSR techniques treat each observation as independent, significance is commonly overestimated, i.e., $p$ is usually <0.0001. While permutation based approaches circumvent this issue for the overall model, they fail to provide estimates of significance of each regressor (i.e., partial effects). One of the recommended methods for obtaining a reasonable F-statistic and related p-value from a functional model is the Likelihood Ratio (LR) test, in which the mean squared error is compared between the original model and a modified model, where the beta estimates are kept constant across the functional domain[39]. This method was employed to test for significance of the overall model.

In addition to reporting the adjusted $R^2$ for the model, we also report the functional $R^2$, which is an adjusted estimate of the variance explained beyond the intercept by comparing the $R^2$ of a model only containing the intercept to a larger model[41]. Functional $R^2$ was calculated for the full model and for each IV (i.e., intercept only compared to intercept plus one regressor). Semi-partial correlations were also calculated using the square root of the difference in $R^2$ between the full model and a model without the each regressor individually. The last step in the FDA was estimating confidence intervals (CIs) for the beta estimates in the PFFR model. Pointwise 95% CIs were calculated using bootstrap resampling of points along the functional response with 1000 iterations. This technique corrects for cases in which the residulas along the functional domain (i.e., density range in this case) are dependent or heteroskedastic.

To summarize our statistical approach, an IV was determined to have a significant association with percolation point for a given density range if: (1) the overall model was significant based on the LR test, (2) the IV was significant in the overall model, and (3) the 95% CIs did not contain 0. This analysis was repeated for all three attack strategies (eigenvector centrality, betweenness centrality, random) both at the whole brain level and within the predetermined language network. Given the high degrees of freedom for modeling, a conservative $p$-value of 0.001 was used for both overall model significance and assessing partial effects of each regressor.

## Reporting summary

Further information on research design is available in the Nature Portfolio Reporting Summary linked to this article.

# Data availability

The percolation point by density data (.mat files) generated in this study, along with minimal linked demographic data, have been deposited in the github repository (https://github.com/willi3by/MEG_FDA). The raw imaging data contain potentially identifiable information, and cannot be shared. Source data are provided with this paper.

# Code availability

Our code has been deposited in a github repository with relevant instructions for reproducing our findings (https://github.com/willi3by/MEG_FDA). The code may need minor modifications (e.g., changing paths to data), but has otherwise been tested and should be immediately ready for use.

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

## Acknowledgements

This study was supported, in part, by grants provided by the Research Institute at Cincinnati Children's Hospital Medical Center [DSK], and the National Institute of Neurological Disorders and Stroke (NINDS) at the National Institutes of Health (NIH; R21NS106631) [DSK/HMG]. We would

like to thank the authors of the *refund package* that was used for Functional Data Analysis. Specifically, we would like to acknowledge Julia Wrobel, Philip Reiss, and Fabian Scheipl for their thoughtful guidance on how to best incorporate their package for our analysis.

## Author contributions

The data used in the current study were acquired by DSK and HMG. BJW and DSK conceptualized the study, carried out data analysis, and drafted the initial verion of the manuscript. All authors were involved in final manuscript preparation and revisions.

## Competing interests

The authors declare no competing interests.
