## [Peer Review File · Nature Communications]

Virtual lesions in MEG reveal increasing vulnerability of the language network from early childhood through adolescenceReviewer #1 (Remarks to the Author):

This study investigated the robustness of the language network from childhood through adolescence using MEG data collected during a story-listening task. The authors simulated the impacts of lesions in the brain network by removing nodes with the highest centrality measures (i.e., eigenvector centrality and betweenness centrality) or by removing the nodes randomly.

By calculating the percolation point (i.e., the fraction of removed nodes for which the size of the second largest connected component is maximum) in various values of threshold on the brain network and then applying the functional data analysis (FDA) approach, the authors found an age-related increase in the language network vulnerability. They also found a gender difference in the vulnerability of the language network. This is a well-written manuscript and an interesting study demonstrating the capability of a simulating lesion approach (or 'In-Silico Attacks' approach as per authors' naming convention) to determine the vulnerability of the language network, with potential clinical and theoretical implications. This study has an incremental novelty by: (a) utilizing a percolation analysis to determine potential vulnerability in language network, while this analysis has already been established and used in other applications; and (b) using FDA for addressing limitations in thresholding of the brain network, while FDA was established in previous studies. Below, please see my other comments.

1. A table for demographic is missing. At least, I expect to see the number and gender of subjects and their age ranges in the four quantiles. Was the gender balanced in 1st and 4th quantiles? Also, the handedness of the subjects should be reported. Any difference in the vulnerability of the language network of the left- and right-handed subjects is interesting to see.
2. The stimuli consisted of 48 stories and 48 noise trials, but it seems that the noise trials were not used, as per methods in Line 158: "The data were then epoched 0-2000ms relative to the onset of each sentence in the stories trials." To remove the non-linguistic components of the auditory stimuli, noise trials should be used. Then the methods in 'MEG Preprocessing and Source Localization' Section should be modified to state how the story vs. noise was contrasted in the LCMV beamformer (e.g., using the common filter approach).
3. Line 148: As per Fig. 1-(b), it seems that the number of nodes was not equal in the right and left hemispheres. Please specify the number of nodes in each hemisphere.
4. Please specify L/R (neurological vs. radiological view) in Figure 1-(a). In Figure 1-(b), nodes for the whole-brain network should also be shown.
5. Line 158: How many artifact ICs, on average across subjects, were rejected for the heart and eye movement artifacts?
6. Line 165-167: Considering an epoch size of 2 seconds (Line 159), the computation of wPLI in very low frequency is questionable. For example, an epoch with a 2 s duration covers only one cycle at 0.5 Hz; thus, the calculation of wPLI at this frequency is unreliable for a 2 s duration trial. In practice, it is recommended to have a couple of cycles in a trial per each frequency for reliable calculation of a phase-based connectivity measure.
7. Line 167: There is no rationale for why the authors generated the adjacency matrix by averaging across all frequencies from 0.5 to 100 Hz. For example, the beta band has been traditionally associated with language processing, and it is interesting to see if the vulnerability of the language network is derived from the connectivity of the network in this band.
8. Line 170, Attack Analyses: This section is hard to follow by a reader who has not been exposed to this type of analysis previously, and I recommend expanding this section to be self-explanatory.
 - For the centrality measures, for instance, you should state that the nodes were sequentially removed from those with the highest centrality.
 - The exact definition of percolation points is unclear. The authors perhaps meant "to identify the percolation point with the fraction of removed nodes for which the size of the second largest connected component is maximum," as previously defined in [Artime, Oriol, et al., Scientific

reports, 2020]. Please make sure to add "fraction of removed nodes" in the definition of percolation point. Please also cite the above reference (or any other relevant papers) here and state that you used the percolation point as defined in the previous study.

9. Lines 179, and 180-188: There is no figure in the manuscript to show the 'mean percolation point' across different thresholds (similar to Fig. 2 but with mean percolation point in the y-axis). Such a figure is informative and should be added to the manuscript. I recommend adding a figure showing the average and STD of the mean percolation point across subjects, separately for two groups (1st and 4th quartile groups).

10. Line 196-197, "Hypothesis testing was performed using 200 permutations": Please specify across which parameter(s) or variable(s) the permutation was performed. In addition, there are 'pointwise 0.05 critical value' and 'maximum 0.05 critical value' in Figs. 2 and 4 that were not defined in Section Functional Data Analysis.

11. Line 206, "Analyses revealed a robust significant effect of age while controlling for sex on percolation point determined by eigenvector centrality (EC) at densities of 6-21% (Fig. 2, top right panel)". This figure shows a non-significant effect for densities above ~17%.

12. Figure 3

- Hubs in younger children are mostly in the non-linguistic areas, e.g., in the occipital area. It is unclear why would occipital areas be the most important vulnerability areas in the whole-brain network during the process of an auditory story listening task.
- It is unclear why the authors select a 2.5% density for this figure while they stated that "The most robust effect occurs around 16% density", and they observed "robust significant effect of age ... at densities of 6-21%".
- Please modify the following sentence in the caption of this figure "our results suggest that the location of hubs decreases as participants get older", as location does not make sense here.

13. Line 272-273, "There was also an overall trend of more critical regions being in the left hemisphere": This is not a fair statement since the number of nodes in the right hemisphere seems to be less than that in the left hemisphere (see Fig. 1-[b]). The ratio of the numbers of critical to total nodes in each hemisphere should be used. The authors should also perform a statistical analysis to investigate whether this ratio was imbalanced in the two hemispheres across subjects.

14. Figure 5: Why did the authors select a 2.5% density while the top-right panel in Fig. 4 does not show a significant effect at this density?

15. Figures 3 and 5: To show the consistency of the results across densities with significant age effects, it is informative to repeat these figures at different (significant) densities. The new figures can be placed in the Supplementary Materials.

Minor comments:

1. Abstract: Please abbreviate MEG
2. Line 128: Please correct "48 nois"
3. Figs 2 and 4: These figures are low resolution in PDF version of the manuscript. A color bar is missing!
4. Line 299 and 317: Please correct the typo "vulnerability"

Reviewer #2 (Remarks to the Author):

The authors evaluated by simulating lesion impact the resilience of brain networks as a function of age in 4-19 year-old children. The analyses were conducted on MEG data recorded from >80 children while they listened to stories by first estimating the functional connectivity across a pre-defined set of nodes across the cortex and by then by removing nodes from the networks thresholded at different densities until the second largest network component peaked in size. This

percolation point that was used as a correlate of network disintegration was evaluated across all possible network densities to quantify the resilience of the whole brain and language networks as a function of age. The study applies novel methodology free of potential biases in such simulated lesion analyses and proposes interesting findings related to the resilience of different brain networks in development. I, however, find that some potential confounding factors were not considered in the analyses and additional analyses would be needed to support the conclusions made in the study. Secondly, as several details of the analysis outcomes are not fully reported, it is presently not possible to evaluate whether different types of lesions would indeed be critical to the different networks.

Specific comments

1. My principal concern related to the analyses and findings is that they do not take into account the differences in field spread properties across differently aged children that potentially confound the MEG-based functional connectivity estimates. The authors apply a functional connectivity metric that is insensitive to direct effects of linear mixing inherent to MEG. However, it is possible that the patterns of how true interactions are detected due to so-called ghost interactions (Palva et al, Neuroimage. 2018; 173:632-643.) are distinct in children from different age-groups. If I understood the methodology correctly, in all children the same number of nodal points (89 and 194 for the language and whole brain network) were used for all children. Thus, in younger children with smaller brain sizes, the nodes are in fact closer to each other than in older children. Accordingly, a true connection between a pair of regions that is detected with MEG also for regions pairs in the vicinity of the true pair is likely to be seen in more "ghost-pairs" in younger children in whom the same vicinity (same spatial distance) includes more possible region pairs that for older children. Using the simulated lesions/in-silico attacks it would then take more node removals to disintegrate the network to the same degree as for older children for whom the true connection is detected in a more restricted set of region pairs. As the use of the same number nodes across the whole subject group is well justified to maintain equal functional division across brain regions within the subject cohort, as far as I see this potential confound cannot be taken into account in the MEG analysis as such. Instead, the authors could evaluate whether there is an independent role of age beyond the brain size in the network resilience. I am not an expert in the use of the Functional Data Analysis so I am not certain how to incorporate the information within the analysis pipeline. But at least in Figure 4 the authors present the results from an analysis on the effects of sex on the percolation point where the subjects' age has been controlled for. It would be beneficial if a similar approach could be used but instead evaluating the effects of age on the percolation point when the brain size is controlled for.

2. I also find that the presented results do not give a clear picture on how the different attack types influence the two networks (whole brain, language). The authors report only the significant observation ($p < 0.05$) showing, e.g., age-related increases in whole-brain vulnerability for eigencentality-based attacks and in language network vulnerability for betweenness-based attacks. However, as the non-significant values for the three attack types (random, eigenvector centrality, betweenness centrality) are not reported, it is not possible to judge whether there indeed are relevant differences between the whole brain and language networks regarding how resilient they are to different types of brain injury. To support the interpretation of the role of provincial versus connector hubs/domain-general versus domain-specific connector, the author should report the full set of results on the in-silico attacks so that the readers are able gain insights into the specificity of certain attack types influencing specific networks instead of all networks.

Reviewer #3 (Remarks to the Author):

General Summary:

The authors' main aim was to understand the effect of in-silico "simulated" attacks on the vulnerability of the whole-brain and pre-defined language network in young and adolescent children. The authors' employed the functional data analysis (FDA) framework to overcome the age-old issue of network thresholding. They used the FDA framework at different network densities and graph theory techniques (namely betweenness centrality and eigenvector centrality) to

evaluate the effect on the network after node removal. The authors were able to prove the pediatric advantage: that the brain recovery after injury is inversely related to age. Overall, this is a novel manuscript employing different modalities (MRI, magnetoencephalography (MEG)) and the combination of the FDA framework and graph theory techniques to understand the effect of insult to the brain and its ability to subsequently recover in pediatric populations.

Abstract:

n/a

Introduction:

The introduction is well-written and appears justified based on prior work.

However, it can be improved by addressing the following suggestions/comments:

1. Please give information about the graph theory measures in brief. A definition as well as a supporting figure will be helpful. The authors talk about it briefly in the discussion, but no context is given before the results. It will be hard for a novice reader to follow the results if the background isn't given.

2. Also, please mention the reason for only and specifically looking at betweenness centrality and eigenvector centrality in the introduction. These should be justified on mathematical and theoretical grounds in the current context.

Methods:

The authors give a good overview of the preprocessing steps.

Here are additional comments:

1. The "pre-defined" language network was made using Neurosyth, a meta-analysis tool. Would you know of any alternate language networks defined in the literature for pediatric populations? If yes, was your analysis approach tested on that?

2. Has this "language network" been validated before within or outside your group?

3. What was the reason for using parcels with 10% activation occupancy? Could that number be lower or higher? How would it affect the spread of the final language regions?

4. "One" language network is defined for a large age-group (ages 4 to 18 years 7 months). Is there information on how the language network develops across age? The authors mention that the language network is bilateral in childhood and becomes more left lateralized and focal with age (line 333-334). Could the analyses the authors ran give different results across the age group as the language network shifts with age?

5. Please give a brief explanation on how the betweenness centrality and eigenvector centrality were computed. Which toolbox was used?

Results:

Here are comments for the results section:

1. Line 212, BC: betweenness centrality as a short form is mentioned for the first time in the manuscript. Please write its full form as well so it's easier to understand.

2. Please show any results with no significance in a supplement.

3. Please share the statistics tables accompanying the results in the results section to better understand the significance results.

4. Please note the beta estimates and significance p-values or F-stat values in the text where it is mentioned that results are significant.

5. Figure 2 and Figure 4: Make the graph size, font and text bigger. It seems fine in the word document, but it seems like the word document was converted to PDF. It was very hard to read the graphs. Zooming in did not help and made the resolution worse (axes of graphs and lines on graph were blurry).

6. Figure 2, top right panel: the maximum 0.05 critical value line isn't plotted. Please show it on the graph.

7. Figure 2, bottom right panel: what does the red dotted red line represent?

8. Figure 3 and 5: Please make a color bar on the side (dark to light) with the frequency information. It will help to understand the color coding faster.

9. Figure 3 and 5: Is the 2.5% density value taken as an example value from the range of densities to show the figures? Please mention it in the manuscript text as well.

Discussion:

1. The overall discussion section is well-written, and is accompanied by a clear explanation for some of their claims given which seem justified based on prior work.

2. The authors do a good job of discussing their results in the domain-general and domain-specific categories, but the links to the specific network measures used here and the attacks could be much more clear. For instance, the authors evoke provincial and connector hubs, but these are not derived from eigenvector or betweenness centrality at face value. In addition, measures like betweenness centrality can potentially have an influence on "information flow" in networks, but information, flow, and the claim that this measure is involved in those constructs is neither explicated nor tested. Overall, this gives the reader the distinct impression that the authors chose two measures of convenience to perform a simulated attack in lieu of surveying and prioritizing network neuroscience more thoroughly and addressing important theoretical gaps.

3. Related to this prevailing problem, the information on what betweenness centrality and eigenvector centrality means is written in the Discussion for the first time. Please move it to the introduction and mention it in more detail in the methods so that there is a better understanding of the concepts and how they tie to the results from the beginning of the manuscript.

Code:

Thank you for submitting the code, the software dependencies, and the instructions to run the code. Also, the github link to the code repository is very helpful.

I had a few comments on code implementation.

1. Initially, I tried running the code on the suggested versions of the software. However, I was unable to install the 'fda' package as it had a dependency of the 'fds' package. Could the authors please report the version of the 'fds' package to install?

2. Also, the 'RColorBrewer' package doesn't have a 1.1.3 version

3. Then, I tried running it on the updated R version and updated package versions. However, it gave me an error.

My comments on implementing it are below:

1. The script was looking for the ages.mat and sex.mat files but in the folder it is called FDA_ages.txt and FDA_sex.txt. Please confirm the code can pull the correct filenames.

2. Line 10: (details of MEG connectivity here....): do details need to be added?

3. I tried running the code on the 'stories_perc_point_FDA_between.mat' file
The code gave an error at line 177 in making the "fRegressList" list variable.

```
[1] "XFDLIST[[ 2 ]] is not an FD or numeric or matrix object."
```

```
[1] "XFDLIST[[ 3 ]] is not an FD or numeric or matrix object."
```

Show Traceback

```
Error in fRegressArgCheck(y, xfdlist, betalists, wt) : An error has been found in either XFDLIST or BETALIST.
```

I couldn't troubleshoot the error so didn't get the code to run completely.

Could you please check the error and make sure the code works on the example data you provided?

4. I also tried running the code on other input data (tried two others) but got the same error at the same line.

Minor:

There are a few grammatical issues in the manuscript. Please correct them.

Some of the instances are noted below:

1. Page 5, Line 98: "parameterized" instead of "parametized"

2. Page 6, Line 128: "noise" instead of "nois"
3. Page 6, Line 139: "language" instead of "langauge"
4. Page 13, Line 299: incorrect spelling of "vulnerability"

We would like to sincerely thank the editor and reviewers for their suggested edits. The resulting manuscript is undoubtedly stronger than the original. Based on these suggestions, we added one covariate and one nuisance regressor to our models. Due to these changes, we had to slightly modify our approach for more flexible modeling and testing of partial correlations. The results remained largely consistent with those of the initial submission. We hope that the responses below and changes to our manuscript adequately address all reviewer concerns.

Reviewer #1 (Remarks to the Author):

This study investigated the robustness of the language network from childhood through adolescence using MEG data collected during a story-listening task. The authors simulated the impacts of lesions in the brain network by removing nodes with the highest centrality measures (i.e., eigenvector centrality and betweenness centrality) or by removing the nodes randomly.

By calculating the percolation point (i.e., the fraction of removed nodes for which the size of the second largest connected component is maximum) in various values of threshold on the brain network and then applying the functional data analysis (FDA) approach, the authors found an age-related increase in the language network vulnerability. They also found a gender difference in the vulnerability of the language network. This is a well-written manuscript and an interesting study demonstrating the capability of a simulating lesion approach (or 'In-Silico Attacks' approach as per authors' naming convention) to determine the vulnerability of the language network, with potential clinical and theoretical implications. This study has an incremental novelty by: (a) utilizing a percolation analysis to determine potential vulnerability in language network, while this analysis has already been established and used in other applications; and (b) using FDA for addressing limitations in thresholding of the brain network, while FDA was established in previous studies. Below, please see my other comments.

1. A table for demographic is missing. At least, I expect to see the number and gender of subjects and their age ranges in the four quantiles. Was the gender balanced in 1st and 4th quantiles? Also, the handedness of the subjects should be reported. Any difference in the vulnerability of the language network of the left- and right-handed subjects is interesting to see.

Thank you, we agree that demographic information should be presented. Table 1 has been added to show sample size, age, sex, and handedness across the 4 age quartiles. We have also updated the models to include handedness as a covariate.

2. The stimuli consisted of 48 stories and 48 noise trials, but it seems that the noise trials were not used, as per methods in Line 158: "The data were then epoched 0-2000ms relative to the onset of each sentence in the stories trials." To remove the non-linguistic components of the auditory stimuli, noise trials should be used. Then the methods in 'MEG Preprocessing and Source Localization' Section should be modified to state how the story vs. noise was contrasted in the LCMV beamformer (e.g., using the common filter approach).

We agree with the reviewer that a stories vs noise contrast is often useful for isolating linguistic components of the neuromagnetic response in functional mapping (e.g., in differential beamforming for presurgical language mapping). However, for connectivity-based analyses, we prefer to focus on the 'active' condition, uncontrasted. We note that use of a stories vs noise contrast assumes that conditions are matched perfectly for 'uninteresting' content – increasingly, this is a difficult argument to make, as we now know that participants do not attend to protracted noise stimuli in the same way as meaningful speech. In studying the neuromagnetic response for stories listening, uncontrasted, we remain sensitive to potentially important connections between auditory and language regions, and the rest of the brain.

The reviewer is correct - the noise data were indeed used in establishing the spatial filter used in source analysis. The inclusion of all trial data stabilizes the covariance estimate, improving beamformer performance.

We have revised the “*MEG Preprocessing and Source Localization*” subsection of the Methods section, (pages 7 and 8), to indicate that a) data were epoched 0-2000ms from the onset of stories and noise trials, b) the stories and noise trial data were concatenated for covariance estimation and construction of a common spatial filter, and c) source activity for the stories listening trials, only, were used in functional connectivity and subsequent attack analyses.

3. Line 148: As per Fig. 1-(b), it seems that the number of nodes was not equal in the right and left hemispheres. Please specify the number of nodes in each hemisphere.

The reviewer is correct that the parcellated language map involves 52 parcels in the left hemisphere, 37 in the right. We have added these details at the end of the “*Defining Whole Brain and Language Networks*” subsection of the Methods (page 7).

4. Please specify L/R (neurological vs. radiological view) in Figure 1-(a). In Figure 1-(b), nodes for the whole-brain network should also be shown.

We apologize for the omission on the uploaded images. Slices are shown in neurological (L=L) convention, to correspond with 3D renderings shown. We have added labeling to indicate laterality on the slices.

5. Line 158: How many artifact ICs, on average across subjects, were rejected for the heart and eye movement artifacts?

Between 0 and 9 (mean 3.82) ICs were identified as stereotypical ocular or cardiac artifact, and rejected. We have added this information under the “*MEG Preprocessing and Source Localization*” subsection of the Methods (page 7).

6. Line 165-167: Considering an epoch size of 2 seconds (Line 159), the computation of wPLI in very low frequency is questionable. For example, an epoch with a 2 s duration covers only one cycle at 0.5 Hz; thus, the calculation of wPLI at this frequency is unreliable for a 2 s duration

trial. In practice, it is recommended to have a couple of cycles in a trial per each frequency for reliable calculation of a phase-based connectivity measure.

We appreciate the reviewer's concern. The Rayleigh frequency for a 2000ms epoch is 0.5Hz ($1/T$, in seconds), reflecting the minimum resolvable frequency in our study. In practice, many labs prefer a low-frequency cut-off corresponding to 1.5, 2, or more cycles, for a given time window – however, increases over the Rayleigh frequency are applied inconsistently, and rarely discussed (in comparison to the counterpart Nyquist), across labs. In our analyses, we apply 2Hz smoothing, so that first frequency bin actually reflects 0.5 to 2.5Hz activity. A 2.5Hz oscillation has a period of 400ms, readily resolved in our epochs.

7. Line 167: There is no rationale for why the authors generated the adjacency matrix by averaging across all frequencies from 0.5 to 100 Hz. For example, the beta band has been traditionally associated with language processing, and it is interesting to see if the vulnerability of the language network is derived from the connectivity of the network in this band.

Thank you for this comment. Increasingly, we observe discordance between connectivity- and oscillatory spectra. In our lab, we observed significant connectivity across canonical bands in children performing verb generation in MEG (e.g., Kadis et al., 2016), through the only reliable oscillatory changes we have seen are specific to alpha and beta, as noted (e.g., Kadis et al., 2011; Sharma et al., 2021). Recently, we showed that outflow and inflow of delta band connectivity corresponds to regions showing low-beta ERD and ERS during verb generation; however, we failed to observe significant effects for beta band connections (Sharma et al., 2022).

We note that most of the recognized oscillatory changes used to map language – alpha and beta event-related desynchrony – reflect spectrally-focused *decreases* in activity. (Alpha and beta ERD are also known to correspond to increases in BOLD signal in fMRI and are likely good assays for 'cortical engagement', in general.) Given decreased spectral power, we might predict decreased alpha and beta band connectivity during language tasks.

We have been studying spectrally-resolved connectivity patterns in a number of populations, including children with reading disability and children born extremely preterm (e.g., Barnes-Davis et al., 2018, 2021), in recent years. Repeatedly, we observe focal connectivity effects that are entirely invisible in broadband or even within canonical bands. As such, we prefer to assess connectivity in narrow frequency steps. Of course, this yields many adjacency matrices, per subject. These can be assessed in multivariate frameworks (e.g., multilayer approach; see Williamson, De Domenico, & Kadis, 2021, Brain Connectivity, for a recent example), or aggregated, as we've done here. We preferred to aggregate using the L2 norm (Euclidean distance), which is relatively sensitive to spectrally-infrequent, strong connections, compared to simple averaging.

We have added these references to the *Functional Connectivity* subsection of the Methods, to motivate our specific approach to assessing connectivity, per subject (page 8):

Recently, we have shown that connectivity patterns differ across frequencies, in children completing language tasks in MEG (Kadis et al., 2016; Sharma et al., 2022); likewise, we have seen spectrally-focused connectivity differences in several patient populations completing language tasks in EEG and MEG (e.g., Barnes-Davis et al., 2018, 2021a, 2021b; Farah et al., 2019). To retain sensitivity to spectrally-focused effects, we assess connectivity in narrow frequency bins, rather than within canonical bands or broadband.

8. Line 170, Attack Analyses: This section is hard to follow by a reader who has not been exposed to this type of analysis previously, and I recommend expanding this section to be self-explanatory.

- For the centrality measures, for instance, you should state that the nodes were sequentially removed from those with the highest centrality.

We have edited this section to the following to address this comment (line 175):

"...betweenness) values in sequential order from those nodes with highest centrality to those with the lowest."

- The exact definition of percolation points is unclear. The authors perhaps meant "to identify the percolation point with the fraction of removed nodes for which the size of the second largest connected component is maximum," as previously defined in [Artime, Oriol, et al., Scientific reports, 2020]. Please make sure to add "fraction of removed nodes" in the definition of percolation point. Please also cite the above reference (or any other relevant papers) here and state that you used the percolation point as defined in the previous study.

We appreciate the reviewer pointing this out. We have changed the text in the manuscript to the following (lines 175-178):

"The percolation point was defined as the fraction of nodes removed for which the size of the second largest connected component peaked in size."

9. Lines 179, and 180-188: There is no figure in the manuscript to show the 'mean percolation point' across different thresholds (similar to Fig. 2 but with mean percolation point in the y-axis). Such a figure is informative and should be added to the manuscript. I recommend adding a figure showing the average and STD of the mean percolation point across subjects, separately for two groups (1st and 4th quartile groups). – Supp Fig

We thank the reviewer for this request and agree that this type of figure is informative. We have added two supplementary figures (Supp Fig 1 and Supp Fig 2), one for each parcellation, that shows the overall mean and standard deviation along with a comparison between the first and fourth age quartiles.

10. Line 196-197, “Hypothesis testing was performed using 200 permutations”: Please specify across which parameter(s) or variable(s) the permutation was performed. In addition, there are 'pointwise 0.05 critical value' and 'maximum 0.05 critical value' in Figs. 2 and 4 that were not defined in Section Functional Data Analysis.

We thank the reviewer for these points. In our updated modeling approach, permutation testing was no longer used to determine F or p-values.

11. Line 206, “Analyses revealed a robust significant effect of age while controlling for sex on percolation point determined by eigenvector centrality (EC) at densities of 6-21% (Fig. 2, top right panel)”. This figure shows a non-significant effect for densities above ~17%.

We thank the reviewer for catching this error. With the revisions, this density range has changed slightly but we have ensured that the density range noted matches the relevant figure (lines 280-297):

“Analyses revealed overall model significance for random attacks ($F = 5.99$, $p < 0.0001$, $R^2 = 0.947$, functional $R^2 = 1.3\%$). There was a significant effect of age while controlling for sex, handedness, and mean node distance on percolation point determined by random attacks at densities below 15% ($F = 31.44$, $p < 0.0001$). Bootstrapped betas for age show a negative relationship (Fig. 2, top panel), where percolation point decreases with age, indicating younger children’s networks are more robust to failure. Results for Random attacks on the whole-brain parcellation are summarized in Figure 2 and Supp Fig 3. There was no significant effect of sex or handedness while controlling for age and mean node distance in this model. Results for random attacks for the whole-brain parcellation are summarized in Figure 2 and Supp Fig 3.

We observed a significant model for the BC-based attack strategy ($F = 6.82$, $p < 0.0001$, $R^2 = 0.934$, functional $R^2 = 8.20\%$). There was a significant effect of age ($F = 126.38$, $p < 0.0001$). Across all densities, bootstrapped betas showed a negative relationship between age and percolation point (Figure 3, top panel). There was no significant effect of sex or handedness while controlling for age and mean node distance in this model. Results for BC-based attacks for the whole-brain parcellation are summarized in Figure 3 and Supp Fig 4. There were no significant effects of age, sex, or handedness on percolation point at any densities for EC based attacks (Supp. Fig. 5).”

12. Figure 3

- Hubs in younger children are mostly in the non-linguistic areas, e.g., in the occipital area. It is unclear why would occipital areas be the most important vulnerability areas in the whole-brain network during the process of an auditory story listening task.

Thank you for this comment. We have added a note in the Results section (line 308) indicating that the younger children tended to have critical hubs clustered in posterior regions, which may reflect relative

reliance on visualization strategies. We have seen this previously, but are cautious about reverse inferencing, here, as we have not studied strategy, experimentally.

- It is unclear why the authors select a 2.5% density for this figure while they stated that “The most robust effect occurs around 16% density”, and they observed “robust significant effect of age ... at densities of 6-21%”.

We thank the reviewer for this comment and agree that the previous density selection did not make sense with the results presented. For our revised figures, we have selected a density of that is consistent with our updated results.

- Please modify the following sentence in the caption of this figure “our results suggest that the location of hubs decreases as participants get older”, as location does not make sense here.

We have modified this caption to correct this error.

13. Line 272-273, “There was also an overall trend of more critical regions being in the left hemisphere”: This is not a fair statement since the number of nodes in the right hemisphere seems to be less than that in the left hemisphere (see Fig. 1-[b]). The ratio of the numbers of critical to total nodes in each hemisphere should be used. The authors should also perform a statistical analysis to investigate whether this ratio was imbalanced in the two hemispheres across subjects.

We agree that this was not a fair statement given the node imbalance in the parcellation. In our updated results, we have removed this point and de-emphasized lateralization.

14. Figure 5: Why did the authors select a 2.5% density while the top-right panel in Fig. 4 does not show a significant effect at this density?

We thank the reviewer for this comment and agree that the previous density selection did not make sense with the results presented. For our revised figures, we have selected a density that is consistent with our updated results.

15. Figures 3 and 5: To show the consistency of the results across densities with significant age effects, it is informative to repeat these figures at different (significant) densities. The new figures can be placed in the Supplementary Materials.

We agree that showing different densities is important for interpretation. Thus, we have added Supplementary Figures 9 and 10 to show the results at 2.5% density above and below (2.5%, 7.5%) our primary results (5%).

Minor comments:

1. Abstract: Please abbreviate MEG

2. Line 128: Please correct “48 nois”
3. Figs 2 and 4: There figures are low resolution in PDF version of the manuscript. A color bar is missing!
4. Line 299 and 317: Please correct the typo “vulnerability” (Now lines 244 and 262)

We thank the reviewer for catching these errors. They have all been addressed in the manuscript.

Reviewer #2 (Remarks to the Author):

The authors evaluated by simulating lesion impact the resilience of brain networks as a function of age in 4-19 year-old children. The analyses were conducted on MEG data recorded from >80 children while they listened to stories by first estimating the functional connectivity across a pre-defined set of nodes across the cortex and by then by removing nodes from the networks thresholded at different densities until the second largest network component peaked in size. This percolation point that was used as a correlate of network disintegration was evaluated across all possible network densities to quantify the resilience of the whole brain and language networks as a function of age. The study applies novel methodology free of potential biases in such simulated lesion analyses and proposes interesting findings related to the resilience of different brain networks in development. I, however, find that some potential confounding factors were not considered in the analyses and additional analyses would be needed to support the conclusions made in the study. Secondly, as several details of the analysis outcomes are not fully reported, it is presently not possible to evaluate whether different types of lesions would indeed be critical to the different networks.

Specific comments

1. My principal concern related to the analyses and findings is that they do not take into account the differences in field spread properties across differently aged children that potentially confound the MEG-based functional connectivity estimates. The authors apply a functional connectivity metric that is insensitive to direct effects of linear mixing inherent to MEG. However, it is possible that the patterns of how true interactions are detected due to so-called ghost interactions (Palva et al, Neuroimage. 2018; 173:632-643.) are distinct in children from different age-groups. If I understood the methodology correctly, in all children the same number of nodal points (89 and 194 for the language and whole brain network) were used for all children. Thus, in younger children with smaller brain sizes, the nodes are in fact closer to each other than in older children. Accordingly, a true connection between a pair of regions that is detected with MEG also for regions pairs in the vicinity of the true pair is likely to be seen in more “ghost-pairs” in younger children in whom the same vicinity (same spatial distance) includes more possible region pairs that for older children. Using the simulated lesions/in-silico attacks it would then take more node removals to disintegrate the network to the same degree as for older children for whom the true connection is detected in a more restricted set of region pairs. As the use of the same number nodes across the whole subject group is well justified to maintain equal functional division across brain regions within the subject cohort, as far as I see

this potential confound cannot be taken into account in the MEG analysis as such. Instead, the authors could evaluate whether there is an independent role of age beyond the brain size in the network resilience. I am not an expert in the use of the Functional Data Analysis so I am not certain how to incorporate the information within the analysis pipeline. But at least in Figure 4 the authors present the results from an analysis on the effects of sex on the percolation point where the subjects' age has been controlled for. It would be beneficial if a similar approach could be used but instead evaluating the effects of age on the percolation point when the brain size is controlled for.

We thank the reviewer for this important comment. We have taken several measures to address this concern. First, mean Euclidean distance between all node pairs was calculated for each participant. As expected, there was a correlation between node distance and age, so we included mean node distance as a nuisance regressor in the model. Fortunately, our model could tolerate this additional predictor (VIF = 2.5). The methods have been updated to reflect these additional steps (lines 193-196):

“To control for potential differences related to field spread effects on small (young) versus larger (adolescent) brains, mean Euclidean distance was calculated for all node pairs and included as a nuisance regressor in all statistical models.”

After reanalyzing with mean node distance as a nuisance regressor, it was clear that the betas were being biased by a few outliers that had high peaks in their percolation by density functions, a phenomenon that was not apparent previously. To account for this, an additional step was added during the FDA to identify these outliers and remove them before final analyses in a data-driven way. This was added to the methods with the following text (lines 214-218):

“First, participants that had a “spike” of percolation point at the lowest density (0.25%), likely due to instability in the percolation point calculation when there are so few nodes, were removed by extracting the last value of the function for all participants, calculating the z-scores for these values, and removing any participant for which this last value had a z-score > 2.”

There was no longer a significant effect of eigenvector centrality in the whole brain analyses. Instead, there was a significant effect over several densities for betweenness centrality at the whole brain level for age. In addition to a significant effect of percolation point for betweenness centrality-based attacks in the stories network, as we previously reported, there was also a significant effect of age with random attacks in both the whole-brain and stories-network. Importantly, all significant effects were in the same direction (decreasing percolation point (i.e., vulnerability) with increasing age). The results section, figures, and relevant sections of the discussion have been updated to reflect these findings.

2. I also find that the presented results do not give a clear picture on how the different attack types influence the two networks (whole brain, language). The authors report only the

significant observation ($p < 0.05$) showing, e.g., age-related increases in whole-brain vulnerability for eigencentrality-based attacks and in language network vulnerability for betweenness-based attacks. However, as the non-significant values for the three attack types (random, eigenvector centrality, betweenness centrality) are not reported, it is not possible to judge whether there indeed are relevant differences between the whole brain and language networks regarding how resilient they are to different types of brain injury. To support the interpretation of the role of provincial versus connector hubs/domain-general versus domain-specific connector, the author should report the full set of results on the in-silico attacks so that the readers are able gain insights into the specificity of certain attack types influencing specific networks instead of all networks.

We thank the reviewer for this request and note that this is an important addition requested by multiple reviewers. We have added 6 supplementary figures (Supp Fig 3-8) that show all results (bootstrapped beta functions with 95% Confidence intervals) for each attack strategy for each parcellation scheme. We hope that this allows readers to fully assess the specificity of these attacks.

Reviewer #3 (Remarks to the Author):

General Summary:

The authors' main aim was to understand the effect of in-silico "simulated" attacks on the vulnerability of the whole-brain and pre-defined language network in young and adolescent children. The authors' employed the functional data analysis (FDA) framework to overcome the age-old issue of network thresholding. They used the FDA framework at different network densities and graph theory techniques (namely betweenness centrality and eigenvector centrality) to evaluate the effect on the network after node removal. The authors were able to prove the pediatric advantage: that the brain recovery after injury is inversely related to age. Overall, this is a novel manuscript employing different modalities (MRI, magnetoencephalography (MEG)) and the combination of the FDA framework and graph theory techniques to understand the effect of insult to the brain and its ability to subsequently recover in pediatric populations.

Abstract:

n/a

Introduction:

The introduction is well-written and appears justified based on prior work. However, it can be improved by addressing the following suggestions/comments:

1. Please give information about the graph theory measures in brief. A definition as well as a supporting figure will be helpful. The authors talk about it briefly in the discussion, but no

context is given before the results. It will be hard for a novice reader to follow the results if the background isn't given.

We thank the reviewer for this important point. We have added a paragraph to the introduction (see response to point 2 below) to address this issue.

2. Also, please mention the reason for only and specifically looking at betweenness centrality and eigenvector centrality in the introduction. These should be justified on mathematical and theoretical grounds in the current context.

Both betweenness centrality and eigenvector centrality (also commonly referred to as eigencentality), are common metrics used to estimate the importance of a node, within a network. Their qualities are distinct, however. The following has been added to the introduction to better explain these metrics (p. 4, lines 75-88):

Two primary strategies can be used when assessing the effects of in silico attacks on a brain network: random and targeted. For targeted attacks, there are several possible metrics that can be used to rank nodes in order of removal. The two strategies used in this work are ranking nodes based on eigenvector centrality (EC) and betweenness centrality (BC), metrics that assess the importance of a node for network functioning¹⁵. EC captures the relative popularity of a node, and can be thought of as an extension of degree, which is simply the number of connections a node possesses. Eigencentality (and related metrics, such as PageRank), consider both the number of connections a node has, as well as the number of connections its topological neighbours possess¹². The advantage of eigenvector centrality over other higher-order metrics of importance based on popularity, is that the metric is derived directly from the eigenvalue of the adjacency matrix, requiring no tuning (cf PageRank, which requires setting 'damping factors' dependent on various assumptions). BC reflects the critical positioning of a node within a network. Nodes with high betweenness centrality serve as bridges, allowing for informational flow to/from other nodes.

Methods:

The authors give a good overview of the preprocessing steps.

Here are additional comments:

1. The "pre-defined" language network was made using Neurosyth, a meta-analysis tool. Would you know of any alternate language networks defined in the literature for pediatric populations? If yes, was your analysis approach tested on that?

We thank the Reviewer for this question. In prior connectivity studies, we have worked extensively with fMRI-derived maps to define the language network. We have used fMRI data from large external cohorts (as in Kadis, Dimitrijevic, Toro-Serey, Smith, & Holland, 2016, Brain Connectivity) and also from the cohorts under investigation (e.g., Barnes-Davis ... & Kadis, 2018, Dev Sci; 2020, Neurolmage Clin; 2020, Sci Rep; 2021, Brain Sciences; 2022, Frontiers Pediatrics).

We see excellent spatial concordance in network definitions across studies, and we feel the map used in the current analysis was highly consistent with our prior work. Here, we prefer a map derived from NeuroSynth, because it's accessible to all, replicable, and unbiased. Studies are included as long as they meet a set of reasonable criteria (outlined in Yarkoni, Poldrack, Nichols, Van Essen, & Wager, 2011, Nat Methods), which have to do with transparency and reporting. We carefully evaluated the first 100 studies (based on loading) that contributed to the resulting NeuroSynth-derived map, to evaluate the diversity of work. We can confirm that studies of *early language learning, dynamic representation in childhood, adulthood and aged cohorts, multilingualism, and patient studies (late talkers, schizophrenia), and the use of various paradigms (including resting state)*, were included.

NeuroSynth can be used to generate a 'uniformity' map, as well as an 'association' map. The former yields a map from studies matching the query term, where the latter yields a map that is selectively related to the search term (i.e., not to other terms). As such, the uniformity map is more extensive (and bilateral). We preferred the uniformity map, to reflect the fact that pediatric language network is known to be relatively extensive and bilateral.

We considered another popular meta-analytic tool, NeuroQuery (neuroquery.org). The map for the search term 'language' is highly concordant with NeuroSynth's association map for the same term.

2. Has this "language network" been validated before within or outside your group?

Please see previous response. In brief, the language uniformity map from NeuroSynth is concordant with our previous fMRI studies of language (various tasks) in childhood.

3. What was the reason for using parcels with 10% activation occupancy? Could that number be lower or higher? How would it affect the spread of the final language regions?

The 10% occupancy criteria is somewhat arbitrary. We recommend using some minimum occupancy criteria to prevent single 'orphan' activations from contributing to final network definition (this is equivalent to setting an activation clustering threshold). In this study, we set a relatively low occupancy threshold, in advance – we preferred the 'inclusive' approach, yielding a language network map that is representative of the NeuroSynth uniformity map, and the extensive pediatric language network.

Depending on the parcellation scheme, a user may wish to adjust their criteria for inclusion. With very fine parcellations (approaching voxel dimensions), the occupancy cutoff would approach 100%.

In all cases, the maps should be visually inspected to confirm representativeness (i.e., Figure 1a can be compared to the map generated by the meta-analytic tool; likewise, the resulting nodal map compared to the 'activation')

4. "One" language network is defined for a large age-group (ages 4 to 18 years 7 months). Is

there information on how the language network develops across age? The authors mention that the language network is bilateral in childhood and becomes more left lateralized and focal with age (line 333-334). Could the analyses the authors ran give different results across the age group as the language network shifts with age?

This is an excellent question, and it's something we struggle with in developmental studies. Indeed, the language network is known to transition from extensive and bilateral to relatively focal and left lateralized, across childhood. However, our attack approach in development requires a common starting point, necessitating a single definition across participants.

Indeed, representation and connectivity are related. With dynamic representation in childhood, the critical components of the network may be shifting. We believe our attack analyses reveal that a subset of hubs become more 'critical' (i.e., removal results in catastrophic dis-integration of the network) with age. We study how the different components of the network interact, here, as changes in representation are relatively well-documented.

5. Please give a brief explanation on how the betweenness centrality and eigenvector centrality were computed. Which toolbox was used?

We have added an explanation of BC and EC in the introduction and added the following line to the methods to specify this point:

"The Brain Connectivity Toolbox (BCT) was used to compute all graph metrics {Rubinov.2010}."

Results:

Here are comments for the results section:

1. Line 212, BC: betweenness centrality as a short form is mentioned for the first time in the manuscript. Please write its full form as well so it's easier to understand.

We thank the review for this point and have added an explanation of BC and its shortform to the introduction.

2. Please show any results with no significance in a supplement.

We agree with the reviewer that this is an important addition and have added Supp Figs 3-8 in response to this comment and comment 2 from reviewer 2.

3. Please share the statistics tables accompanying the results in the results section to better understand the significance results.

We thank the reviewer for this important request and have added statistics tables to all relevant figures.

4. Please note the beta estimates and significance p-values or F-stat values in the text where it is mentioned that results are significant.

As with the previous related comment, we thank the reviewer for this important request and have added the relevant statistics to the text for better understanding of the results.

5. Figure 2 and Figure 4: Make the graph size, font and text bigger. It seems fine in the word document, but it seems like the word document was converted to PDF. It was very hard to read the graphs. Zooming in did not help and made the resolution worse (axes of graphs and lines on graph were blurry).

We appreciate the reviewer's concern about the figure resolution. For the final submission, figures will be submitted separately as high-resolution TIFF images, which will give us more flexibility to adjust these features.

6-9; 6. Figure 2, top right panel: the maximum 0.05 critical value line isn't plotted. Please show it on the graph. 7. Figure 2, bottom right panel: what does the red dotted red line represent? 8. Figure 3 and 5: Please make a color bar on the side (dark to light) with the frequency information It will help to understand the color coding faster. 9. Figure 3 and 5: Is the 2.5% density value taken as an example value from the range of densities to show the figures? Please mention it in the manuscript text as well.

We thank the reviewer for these thorough points. In the revised figures, figure captions, and results, we have considered and checked points 6-9 to ensure that these points have been addressed.

Discussion:

1. The overall discussion section is well-written and is accompanied by a clear explanation for some of their claims given which seem justified based on prior work.

We thank the reviewer for this kind remark!

2. The authors do a good job of discussing their results in the domain-general and domain-specific categories, but the links to the specific network measures used here and the attacks could be much more clear. For instance, the authors evoke provincial and connector hubs, but these are not derived from eigenvector or betweenness centrality at face value. In addition, measures like betweenness centrality can potentially have an influence on "information flow" in networks, but information, flow, and the claim that this measure is involved in those constructs is neither explicated nor tested. Overall, this gives the reader the distinct impression that the authors chose two measures of convenience to perform a simulated attack in lieu of

surveying and prioritizing network neuroscience more thoroughly and addressing important theoretical gaps.

We appreciate the reviewer's concern and have added some necessary background to the introduction (p. 4, lines 75-88), motivating our choice of studying eigenvector and betweenness centrality-based attacks. We have also expanded the discussion (p.13, lines 334-353) to address the reviewer's concerns. We hope that the relationship between these centrality measures and domain-general vs domain-specific hubs is clearer now.

3. Related to this prevailing problem, the information on what betweenness centrality and eigenvector centrality means is written in the Discussion for the first time. Please move it to the introduction and mention it in more detail in the methods so that there is a better understanding of the concepts and how they tie to the results from the beginning of the manuscript.

We thank the reviewer for this comment. We have moved this information to the introduction and hope our explanation there makes for a better understanding (see response in the introduction comments, point 2).

Code:

Thank you for submitting the code, the software dependencies, and the instructions to run the code. Also, the github link to the code repository is very helpful.

I had a few comments on code implementation.

1. Initially, I tried running the code on the suggested versions of the software. However, I was unable to install the 'fda' package as it had a dependency of the 'fds' package. Could the authors please report the version of the 'fds' package to install?

This has been added to the README.md file.

2. Also, the 'RColorBrewer' package doesn't have a 1.1.3 version

With the updated code, it is no longer necessary to specifically include this package so version should not affect results.

3. Then, I tried running it on the updated R version and updated package versions. However, it gave me an error.

We apologize for this issue. With the updated code and dependencies, everything should work smoothly. If there are any issues, please feel free to reach out either on GitHub or by email.

My comments on implementing it are below:

1. The script was looking for the ages.mat and sex.mat files but in the folder it is called FDA_ages.txt and FDA_sex.txt. Please confirm the code can pull the correct filenames.

We apologize for this error. To make things easier, all demographic data has now been aggregated in an R dataframe (demos.R) that is included in the study folder. Code to load this has also been added to the Rmd main analysis file.

2. Line 10: (details of MEG connectivity here....): do details need to be added?

We thank the reviewer for catching this error. More details have now been added to the Intro of the Rmd file.

3. I tried running the code on the 'stories_perc_point_FDA_between.mat' file
The code gave an error at line 177 in making the "fRegressList" list variable.

```
[1] "XFDLIST[[ 2 ]] is not an FD or numeric or matrix object."
```

```
[1] "XFDLIST[[ 3 ]] is not an FD or numeric or matrix object."
```

Show Traceback

```
Error in fRegressArgCheck(y, xfdlist, betalist, wt) : An error has been found in either XFDLIST or BETALIST.
```

I couldn't troubleshoot the error so didn't get the code to run completely.

Could you please check the error and make sure the code works on the example data you provided?

We again apologize for this inconvenience. In the updated analysis, fRegress is no longer needed for the primary statistical modeling. Modeling and data input is now much less involved so should be more straightforward to troubleshoot if there are initial errors. We have tested on all 3 operating systems with successful output.

4. I also tried running the code on other input data (tried two others) but got the same error at the same line.

Please see response to point 3.

Minor:

There are a few grammatical issues in the manuscript. Please correct them.

Some of the instances are noted below:

1. Page 5, Line 98: "parameterized" instead of "parametized"
2. Page 6, Line 128: "noise" instead of "nois"
3. Page 6, Line 139: "language" instead of "langauge"
4. Page 13, Line 299: incorrect spelling of "vulnerability"

We thank the reviewer for catching these errors and have corrected them in the revised manuscript.

Reviewer #1 (Remarks to the Author):

Thanks to the authors for carefully revising the manuscript and responding to my comments. While there still exist certain points that warrant further discussion, I find myself aligned with the authors' overarching efforts in refining the manuscript and addressing the raised concerns.

I do, however, wish to highlight an issue related to Supplementary Figures 1 and 2. The subplot titles within these figures remain somewhat perplexing, and unfortunately, the figure captions themselves do not adequately alleviate this confusion. I have observed that the titles of subplots in the top row mirror those in the bottom row. Furthermore, the titles of the top subplots refer to a "Percolation Point Difference Between First and Fourth age Quartiles." However, the visual representations do not present the indicated 'Difference,' and in addition, two plots in distinct red and blue colors correspond to the two quartiles.

Furthermore, I would like to note that the resolution of the Supplementary figures appears to be insufficient, which may hinder their comprehensibility.

Reviewer #2 (Remarks to the Author):

The authors have conducted very thorough new analyses and markedly extended the manuscript related to several key aspects. These changes and the answers given by the authors have addressed all my previous concerns.

Reviewer #3 (Remarks to the Author):

The authors addressed all the comments with detailed explanations that significantly improved the manuscript. The additional text and figures have resolved my concerns.

I also appreciated the improved code repository. I was able to run the code.

I have no further comments at this time and appreciate the authors' additional work.

Review Responses

Reviewer #1 (Remarks to the Author):

Thanks to the authors for carefully revising the manuscript and responding to my comments. While there still exist certain points that warrant further discussion, I find myself aligned with the authors' overarching efforts in refining the manuscript and addressing the raised concerns.

We thank the reviewer for this kind comment and have addressed the remaining concerns below.

I do, however, wish to highlight an issue related to Supplementary Figures 1 and 2. The subplot titles within these figures remain somewhat perplexing, and unfortunately, the figure captions themselves do not adequately alleviate this confusion. I have observed that the titles of subplots in the top row mirror those in the bottom row. Furthermore, the titles of the top subplots refer to a "Percolation Point Difference Between First and Fourth age Quartiles." However, the visual representations do not present the indicated 'Difference,' and in addition, two plots in distinct red and blue colors correspond to the two quartiles.

Furthermore, I would like to note that the resolution of the Supplementary figures appears to be insufficient, which may hinder their comprehensibility.

We thank the reviewer for pointing out the lack of clarity with these supplementary figures. We have adjusted the figure titles and captions to clear up confusion about what is being represented (not the difference, just each group separately with standard deviation). Also, we have done our best to improve the resolution of the images in the final supplementary pdf.

Reviewer #2 (Remarks to the Author):

The authors have conducted very thorough new analyses and markedly extended the manuscript related to several key aspects. These changes and the answers given by the authors have addressed all my previous concerns.

We thank the reviewer for the kind comments and appreciate their time in thoroughly reviewing our work!

Reviewer #3 (Remarks to the Author):

The authors addressed all the comments with detailed explanations that significantly improved the manuscript. The additional text and figures have resolved my concerns.

I also appreciated the improved code repository. I was able to run the code.

I have no further comments at this time and appreciate the authors' additional work.

We thank the reviewer for their recommendations that undoubtedly improved the manuscript and for testing the code to make sure everything works as expected!